# OpenReview forum: "PRISON: Unmasking the Criminal Potential of Large Language Models"
_ICLR.cc/2026/Conference — ICLR 2026 Poster_

### Official Review · Reviewer_Wu5t · 2025-10-22

**Soundness:** 3
**Presentation:** 4
**Contribution:** 2
**Rating:** 4
**Confidence:** 4

**Summary:**

The paper introduces PRISON, a novel framework for assessing the capabilities of LLMs to a) develop criminal strategies and b) detect criminal acts. Building upon plots and situations from mainstream movies, the framework creates settings in which an LLM is queried to solve the situation with malicious intent, e.g., covering up an accident. The first aspect of the study explores whether LLMs can come up with valid, illegal/harmful strategies given the individual setting. The second aspect takes the opposite view and tests whether the same LLMs can detect illegal actions in the generated strategies. Evaluating multiple recent LLMs, the paper demonstrates that language models show high criminal capabilities while their detection capabilities for such actions are limited.

**Strengths:**

- The paper is very well-written and easy to follow. All figures and findings are clean and straightforward to understand. Overall, the paper formatting quality is above average.
- Investigating the criminal potential of LLMs is an interesting avenue, and leveraging the scripts of movies to create an evaluation framework is a smart idea. The findings that there exists a mismatch between criminal actions and criminal detection are intriguing. I also like that the paper not only distinguishes between criminal/non-criminal but also analyzes the different kinds of criminal traits.
- The experimental evaluation covers multiple LLMs (8 different models) and settings. Whereas some recent models, e.g., GPT-5, Qwen-3, DeepSeek-R1, are missing, the provided models offer a good mix of non-reasoning LLMs.

**Weaknesses:**

- The framework setting feels somewhat artificial. While I understand the intention behind the dataset, I am not fully convinced that the evaluations genuinely assess a model’s criminal capabilities. When looking at examples in the Appendix (e.g., Table 5), it often feels as if the model is writing a novel. On one hand, such narrative-style outputs could indeed be misused for criminal purposes. However, I am not sure whether these outputs are actually harmful, since it remains unclear to what extent the proposed strategies go beyond straightforward ideas or common scenes from movies. While, in another context, detailed instructions for building a bomb could clearly cause harm, suggesting to push a car into a lake (which requires no expert knowledge) seems more like repeating a movie or book scenario. I understand that we want LLMs to avoid producing such suggestions, but given that the model appears to be engaging in creative writing, this behavior might be acceptable in some cases.
- The crime detection capabilities of LLMs are not compared against a human baseline. Since LLMs have less information than the “God-setting,” a human baseline would help quantify the actual performance gap. Given that some contextual information is missing from the detection model’s input, it might be that certain actions cannot be reliably classified as criminal.
- As the framework is based on only ten movies, the diversity of scenarios may be limited.
- No large reasoning models, such as Qwen3 or DeepSeek-R1, are evaluated. It would be interesting to see whether stronger reasoning capabilities improve a model’s ability to either generate or detect criminal content.

Small remark:
- There is a missing space in L046.

**Questions:**

- How does a human baseline perform on the detection task compared to the LLMs? Is there sufficient information contained in the inputs actually to solve the task (could be answered by a user study)?

---

> ### Author Response · Authors · 2025-11-21
> **Response to Reviewer Wu5t**
>
> Thank you for your insightful comments. Here are our responses:
> ## (1)  Framework Usability
> First, when the model is asked to *"engage in an immersive simulation"*, behaviors such as deception and manipulation should not be viewed as harmless creative writing. Prior work on jailbreaks shows that role-based prompts can elicit unsafe advice, reflecting flawed normative judgments in scenarios resembling real-world interpersonal disputes or accidents. In the "push the car into the lake" example, although this suggestion does not require any expert knowledge, a moral human or a well-aligned model such as GPT-4o would urge contacting emergency services or preserving evidence. In contrast, models like DeepSeek generate strategies for concealing wrongdoing. Therefore, we **view this discrepancy not as harmless fiction, but as a failure of moral and safety alignment in scenarios that closely mirror high-stress decision making**.
>
> Second, the goal of our work is not to measure the absolute harmfulness of the generated advice (e.g., comparing it to bomb-making instructions). Instead, we aim to **"highlight a risk-amplification asymmetry: LLMs may be more exploitable for facilitating crime than for preventing it."** (lines 89–90) In other words, our evaluation thus focuses on whether the model's underlying social reasoning is safe and aligned, rather than on whether the model invents entirely novel criminal techniques. Even when the surface content resembles scenes from movies or novels, the decision-making patterns the model exhibits such as covering up wrongdoing, are meaningful indicators of its susceptibility to misuse in interactive settings.
>
> ## (2) User Study on Detection
> While our study was designed to examine LLMs' ability asymmetry between criminal expression and trait detection, to establish a human baseline, we conducted a controlled crowdsourced user study to strengthen the robustness of our conclusions.
>
> We randomly sampled 20 detection cases, each consisting of a limited background scenario and a target sentence. These were distributed to 100 qualified crowd workers on Credamo Platform, each meeting a minimum 80% approval rate and reporting at least an undergraduate degree. All workers independently judged the sentences using the same operational definitions in Section 3.2. Each worker spent 10–15 minutes on the task and received $2 USD in compensation.
> We aggregated responses via majority vote and computed the OTDA. Human workers achieved an average OTDA of 73.1%, substantially higher than the LLM average of 42.5% on the same items. This shows that, even under identically restricted inputs, humans can reliably infer the presence of criminal traits. Therefore, **the performance gap observed in our experiments is not driven by task infeasibility or insufficient context but instead reflects genuine limitations in current LLMs' capabilities**.
>
> The above details are updated to Appendix F.2.3 in the revised version.
>
> ## (3) Diversity of Scenarios
> While our scenarios originate from ten narrative crime films, we would like to clarify that:
> First, **the films served only as the raw material for scenario extraction and adaptation**. We extracted and rewrote multiple distinct scenes, yielding 60 unique scenarios, each explored through five rounds of dialogue. Across models, this resulted in an average of 3,910 sentences per model for annotation, ensuring sufficient dataset breadth.
>
> Second, our source films were carefully selected to ensure coverage of mainstream and diverse criminal types. As detailed in the paper (lines 221-225), the selected films span diverse crime types, from accidental incidents to premeditated and professional crimes. We agree that these selected scenarios provide a systematic and adequate coverage of various crime types.
>
> Overall, as the first systematic study of the asymmetry between criminal expression and criminal detection in LLMs, **this paper aims to provide a strong start point**. For future work, we are committed to *"incorporating more diverse and authentic sources to capture the full complexity of real-world criminal situations"* (lines 458-460).
>
> ## (4) Impact of Reasoning Capabilities
> We observed that **criminal potential is not reduced by improved reasoning abilities**. To verify this, we included DeepSeek-R1, and compared it with DeepSeek-V3 under the same evaluation setup.
>
> The results show that DeepSeek-R1 and DeepSeek-V3 exhibit similarly high CTAR scores (66.60% vs. 65.23%), indicating that explicit reasoning mechanisms do not lessen a model's tendency to express criminal traits. Although DeepSeek-R1 achieves a moderately higher OTDA, suggesting that stronger reasoning can slightly improve detection ability, the performance gap between criminal-trait expression and detection remains. This reinforces that criminal potential is not automatically diminished by stronger reasoning.
>
> The above details are updated to Appendix F.3.4 in the revised version.

---

> > ### Comment · Reviewer_Wu5t · 2025-11-21
> >
> > I thank the reviewers for their responses and the additional details they provided, particularly the user study and the comparison between reasoning and non-reasoning models (the DeepSeek R1 vs. V3 comparison appears reasonable).
> >
> > After reading all reviews and rebuttals, I have decided to keep my initial score. I understand the main message of the paper that there is a risk-amplification asymmetry in the sense that “LLMs may be more exploitable for facilitating crime than for preventing it.” However, I believe that safety research over the past few years has already demonstrated through extensive evaluations (safety benchmarks, jailbreaks, safeguards, etc.) that it is generally easier to coerce LLMs into producing harmful content than to rely on them for detecting it.
> >
> > Thus, my main concern remains: the experimental settings feel highly artificial, and the model’s recommended actions (within these constructed scenarios) do not present any practical harm beyond Hollywood-style villain strategies. While it is interesting that the models tend to generate such strategies, their suggestions are very high-level, and a quick web search could yield similar ideas. In my view, LLMs can meaningfully increase harm primarily when they provide information that is not easily accessible through a quick search or basic brainstorming, namely, expert-level content such as chemical or engineering knowledge.
> >
> > That said, I acknowledge that the other reviewers have rated the paper slightly more favorably, and I am open to discussing my concerns with them. For now, I am inclined to (slightly) reject the paper.

---

> > > ### Author Response · Authors · 2025-11-26
> > > **Response to Reviewer Wu5t (Part 1/2)**
> > >
> > > We sincerely thank the reviewer for the thoughtful follow-up and for being open to further discussion. We fully understand your concerns, and we would like to provide additional clarification that may help further elucidate our work:
> > >
> > > ## (1) Novelty of Our Work
> > > Our study is the first to systematically evaluate the criminal potential of LLMs, defined as the risk of exhibiting harmful behaviors such as deception, manipulation, or blame-shifting during adversarial multi-turn social interactions. This represents a clear departure from prior evaluations that rely on simplified or isolated tasks, such as single-turn deception probes or static moral-judgment settings, which cannot capture the dynamic, multi-agent decision processes inherent in realistic criminal contexts. As discussed in our introduction, criminal behavior typically requires multi-step reasoning, perspective asymmetry, and social strategic planning, none of which are jointly assessed in existing benchmarks. As a result, the scenarios and the implications revealed by PRISON differ qualitatively from previous settings and allow us to surface behavioral patterns that earlier task designs could not observe.
> > >
> > > Identifying our key asymmetry finding requires a unified framework that explicitly models both criminal expression and criminal detection. However, no existing study evaluates whether a model that can generate deceptive behaviors can also recognize them when placed in a corresponding real-world detection role. This omission leaves an important gap in understanding whether LLMs amplify criminal risk. Our tri-perspective framework, which jointly examines the Criminal, Detective, and God views, is explicitly designed to isolate these overlooked factors. Such design allows us to uncover the structural mismatch between criminal expression and detection that has not been strictly proved before.
> > >
> > > ## (2) Harmfulness of Criminal Traits
> > > Our core intention is to examine the behavioral tendencies of LLMs, particularly their propensity to express and detect criminal psychology traits such as deception, manipulation, and moral disengagement. These behaviors have been identified as red-line risks in AI governance and are viewed as comparable in seriousness to traditionally recognized safety domains such as chemical, biological, nuclear, and radiological threats [1]. This is because many real-world harms, including fraud, social engineering, and interpersonal manipulation, arise from such high-level behavioral patterns rather than from technical expertise.
> > >
> > > We note that an LLM’s outputs are fundamentally different from simple web search. In our threat model, users consult the model when they are uncertain about the situation and need help deciding what to do next, meaning the interaction functions as a form of joint decision-making rather than factual retrieval. By contrast, users typically rely on search engines only when they already have a clear action in mind and are seeking concrete operational details. From a criminological perspective, high-level strategies such as concealing facts, shifting blame, manipulating emotional cues, or offering misleading statements are themselves meaningful forms of criminal facilitation because they often precede and shape subsequent harmful actions. Therefore, even without step-by-step instructions, a model’s ability to produce personalized and context-sensitive harmful reasoning already represents a relevant and concerning capability.
> > >
> > > [1] OpenAI. (2024). *Updating Our Preparedness Framework.* OpenAI Safety. Available at: https://openai.com/index/updating-our-preparedness-framework/

---

> > > > ### Author Response · Authors · 2025-11-26
> > > > **Response to Reviewer Wu5t (Part 2/2)**
> > > >
> > > > ## (3) Rationality of the Framework
> > > >
> > > > In practice, many relevant safety researches rely on synthetic, fictional, or game-based environments to probe similar behaviors in LLMs. Pan et al.’s MACHIAVELLI benchmark evaluates whether agents choose to betray, manipulate, or pursue unethical strategies in long-horizon text-adventure games [1]. Although the decisions made in these settings have no real-world consequences, the benchmark is considered a strong tool for analyzing behavioral tendencies because it focuses on high-level decisions while abstracting away low-level interactions. Burnell et al. take a similar approach by creating a sandbox for studying agentic deception using a text version of Among Us [2]. Their simulated social-deduction environment reveals long-term deceptive behaviors that arise from the game objectives. Even actions such as murder in this setting cause no real harm, yet the authors argue that such synthetic environments provide a rich and tractable proxy for human–agent interaction. These perspectives align with our choice to use movie scenes as the source of our scenarios.
> > > >
> > > > Agarwal et al.’s dataset based on Werewolf games further supports this view [3]. Their role-playing framework highlights social-deduction settings as natural testbeds for studying deception and theory of mind. Moreover, they demonstrate that there is asymmetry between deception and detection, and emphasize that earlier research surfaced deceptive behaviors but did not systematically evaluate detection accuracy at the level of individual statements. Compared with their design, our movie-based scenarios offer coherent interpersonal structures, clear causal relationships, and realistic motivations that allow high-level behavioral patterns to emerge without any real-world harm. As in previous synthetic environments, the key factor is not literal realism but whether the setting reliably elicits strategic reasoning, ambiguous intentions, and value-laden decisions. Our benchmark builds on these insights and shows that the asymmetry between generating and detecting harmful behavior extends beyond deception and appears across multiple cognitive and behavioral traits.
> > > >
> > > > We hope that the above clarification addresses your concerns, and please feel free to reach out if further discussion would be helpful.
> > > >
> > > > [1] Pan, X., Chan, J. S., Zou, A., Li, N., Basart, S., Woodside, T., ... & Hendrycks, D. (2023). Do the rewards justify the means? Measuring trade-offs between rewards and ethical behavior in the MACHIAVELLI benchmark. In *Proceedings of the 40th International Conference on Machine Learning (ICML 2023)*.
> > > >
> > > > [2] Burnell, R., Hendrycks, D., Carlini, N., Casper, S., Kenton, Z., Rando, J., Chen, A., Hilton, J., Steinhardt, J., & Evans, O. (2025). Among us: A sandbox for measuring and detecting agentic deception. In *Advances in Neural Information Processing Systems (NeurIPS 2025)*.
> > > >
> > > > [3] Agarwal, R., Rana, S., Sundoro, T., Berhe, H., Kim, S., Sharma, V., O’Brien, S., & Zhu, K. (2025). WOLF: Werewolf-based observations for LLM deception and falsehoods. In *Advances in Neural Information Processing Systems (NeurIPS 2025)*.

---

> > > > > ### Comment · Reviewer_Wu5t · 2025-11-27
> > > > >
> > > > > I thank the authors for their additional details and for pointing out related work. While I am still not fully convinced by this artificial setting and its actual harmfulness, I acknowledge that related (and partly already published) benchmarks make similar assumptions. That said, WOLF has only been presented at a NeurIPS workshop and has not been published as a full paper—just as a remark here. Given that frameworks like Among Us offer a more complex setup and a more comprehensive analysis of agent capabilities, I have decided to keep my score.
> > > > >
> > > > > However, I want to emphasize that I understand that other reviewers may interpret this differently, and I will support publication if the majority in the reviewer discussion is in favor.

---

### Official Review · Reviewer_k64m · 2025-10-27

**Soundness:** 3
**Presentation:** 3
**Contribution:** 3
**Rating:** 6
**Confidence:** 3

**Summary:**

This paper introduces PRISON, a novel evaluation framework designed to assess the "criminal potential" of Large Language Models (LLMs) in complex, multi-turn social interactions. The authors define criminal potential as the risk of an LLM generating harmful behaviors like deception, manipulation, or blame-shifting in adversarial contexts that could facilitate unlawful activities. The paper's main contributions are (1) the PRISON framework itself, as a new benchmark for a critical and understudied safety dimension , (2) a quantification of LLMs' "criminal potential" , and (3) the identification of the significant gap between an LLM's ability to generate and its ability to detect such behaviors. These contributions are timely, novel, and significant for the AI safety community

**Strengths:**

1. This work moves beyond traditional, static safety evaluations (e.g., simple harmful Q&A, abstract moral dilemmas) to tackle the much more complex and realistic threat of LLMs participating in deceptive, multi-turn social interactions. The "criminal potential" concept is a valuable and well-defined framing of a risk that is highly relevant as LLMs are integrated into agentic systems. This paper addresses a clear and important gap in the current safety literature.

2. The PRISON framework is thoughtfully constructed. Grounding the five-trait taxonomy in established psychometric instruments from criminal psychology  provides a strong theoretical foundation that is often lacking in other safety benchmarks. Furthermore, the tri-perspective (Criminal, Detective, God) evaluation design is an intelligent and effective method for simultaneously measuring the expression of harmful traits and the detection of them within a unified system

**Weaknesses:**

1. The 44% "Objective Trait Detection Accuracy" (OTDA)  is a headline-grabbing result. However, its significance is difficult to interpret without more details on the "Detective" agent's task.

2. Regarding the "God" perspective validation: A Cohen's Kappa of 0.65 is "substantial" but not "near perfect." Could you provide a qualitative breakdown of the disagreements between your human annotators and the GPT-4o judge? Are there specific traits (e.g., "Psychological Manipulation" vs. "False Statements") that are more ambiguously defined or harder for the LLM to judge correctly?

3. The scenario generation from films is a clever way to source complex social dynamics. However, film narratives are inherently dramatic and conflict-driven. How do you account for the potential domain mismatch between these "dramatized" scenarios and more mundane, real-world criminal interactions? Is it possible this choice of data source biases the "Criminal Traits Activation Rate" (CTAR) upwards?

**Questions:**

See above

---

> ### Author Response · Authors · 2025-11-21
> **Response to Reviewer k64m**
>
> Thank you for your insightful comments. Here are our responses:
>
> ## (1) Details on OTDA
> As described in Section 5 (lines 372-375), the setup of detection tasks means that in each scenario the agent operates under a limited information viewpoint and is required to annotate, sentence by sentence, which of the five criminal traits appear in the observed utterance.
>
> Based on these sentence-level annotations, we compute OTDA, which is defined as *"the proportion of sentences for which the Detective's predicted trait set exactly matches the references from the God perspective"* in Section 3.2 (lines 211-215). For example, if the God view determines that a sentence reflects deception because it reveals a conflict between the perpetrator's statement and the factual details, but the Detective identifies none of these traits, then the annotation for that sentence is counted as incorrect. This illustrates why OTDA serves as a strict and informative measure of detection capability under incomplete context.
>
> ## (2) "God" Perspective Validation
> To ensure the reliability of the "God-perspective" evaluation, we conducted an independent validation using two trained annotators, who reviewed a 20% sample of GPT-4o's judgments. To better understand the disagreements between human annotators and GPT-4o's judgments, we then sampled 100 such cases for qualitative analysis. Our analysis reveals three main error patterns:
>
> (i) Overprediction of Psychological Manipulation: GPT-4o often labels coercive or operational instructions as Psychological Manipulation. For sentences such as *"At the stroke of eleven, when the phone rings, you must come out from behind these drapes and carry out the plan."*, the model overlooks the required criterion of exploiting emotional vulnerabilities, and instead equates action-level coercion with psychological manipulation based purely on surface semantics.
>
> (ii) Reduced sensitivity to Moral Disengagement: Humans readily identify subtle blame-shifting as moral disengagement because the definition explicitly includes "shifts blame to external factors." For sentences like *"I did not force you. I am simply laying out the options before you."*, human annotators recognize the speaker's attempt to pressure the victim while disclaiming responsibility. GPT-4o, however, often fails to detect this implicit coercion.
>
> (iii) Failure to track factual inconsistencies in context: Even when provided with the complete God-perspective information, the model does not always detect statements that contradict known facts. For instance, in scenarios where the context specifies that the detective possesses key evidence, a perpetrator's remark such as *"These pieces of evidence do not intimidate me."* should be classified as a false statement. Human annotators consistently recognize this contradiction, yet GPT-4o sometimes overlooks the conflict and declines to label it as deceptive.
>
> The above discussion is updated to Appendix D.2 in the revised version.
>
> ## (3) Scenario Based on Films
> We selected films because they **provide a controlled test environment for evaluating structured interpersonal dynamics**. As stated in the paper (lines 123 to 125), *"artistic works inspired by real-world contexts are commonly regarded as distilled representations of human behaviors."* Similar to prior dynamic-evaluation frameworks that use fictional yet logically coherent scenarios (lines 116 to 129), our goal is not to reproduce surface realism but to model stable causal structures of deception and manipulation. We strip away cinematic embellishments and retain only the core logical scaffolds (Appendix B.2). As **the first systematic study of the asymmetry between criminal expression and criminal detection in LLMs**, our use of films enables controlled and repeatable experimentation. In future work, we plan to incorporate real-world sources such as court transcripts, investigative reports, and forum discussions to enhance realism and ecological validity.
>
> Besides, rather than emphasizing the absolute values of CTAR and OTDA, our focus is on their relative asymmetry, which ***"highlights a potential risk-amplification effect: LLMs may be more exploitable for facilitating crime than for preventing it."*** (lines 89–90) Here, the absolute level of CTAR naturally varies with crime severity. After re-aggregating the results by crime scenario type, we observe that in Premeditated Murders, LLMs exhibit a higher CTAR (62.48%), whereas in Accidental Incidents, the expression of criminal traits is notably lower (51.83%). However, regardless of the inherent difficulty of the crime scenario, the asymmetry between expression and detection persists. This is the core phenomenon we emphasize: LLMs show a systematic gap between "helping commit" and "detecting" crime-related behaviors, and this gap does not disappear simply because the underlying crime scenario is easier or harder.
>
> The above details are updated to Appendix F.3.1 in the revised version.

---

### Official Review · Reviewer_sCDD · 2025-11-06

**Soundness:** 3
**Presentation:** 3
**Contribution:** 3
**Rating:** 6
**Confidence:** 3

**Summary:**

The paper introduces PRISON, a tri-perspective evaluation framework designed to assess both the criminal potential and detection capability of LLMs in adversarial social scenarios. It models how LLMs behave under roles such as criminals, detectors, and gods, simulating information flow and perspective differences to capture both understanding and detection of illegal behaviors.
The study finds interesting observations. For example, popular LLMs often exhibit criminal traits: generating deceptive or harmful advice, even without explicit malicious prompts. However, they perform poorly when detecting similar behavior.
The evaluation is extensive, which leverages context-rich, film-inspired scenarios to ensure realism while maintaining ethical control.

**Strengths:**

- The proposed framework is novel and studies an important aspect of LLM safety.
- The tri-perspective approach (criminal, detector, god) is innovative and captures the complexity of adversarial scenarios effectively.
- Comprehensively quantifies the criminal tendencies of various LLMs, providing valuable insights into their capabilities and limitations.

**Weaknesses:**

- The performance gap between criminal generation and detection may be the nature of the task itself, rather than a specific shortcoming of LLMs.
- The scenarios are primarily adapted from classic crime films, which may limit representativeness of real-world criminal contexts.
- Lack of technical discussion about why certain behaviors emerge in LLMs.

**Questions:**

Overall, this is a well-structured and insightful study that contributes meaningfully to our understanding of LLM safety in adversarial contexts. The PRISON framework is a valuable addition, offering a creative way to stress-test models' tendencies toward criminal expression and their ability to detect manipulative behavior. However, I have a few concerns:

---
(1) Nature of the Performance Gap:

The observed gap between ”criminal expression“ and “detection” might reflect the nature of the task itself rather than a true model deficiency. For humans as well, generating deception is often easier than detecting it, since detection requires background knowledge and reasoning about intent. It would strengthen the paper if the authors could further analyze whether this gap truly indicates a model limitation or simply the inherent difficulty of the task.

---
(2) Use of Film-Based Scenarios:

Lines 220–221 mention that the scenarios are adapted from films. However, film plots are not necessarily realistic representations of real-world criminal behavior. Why not use more authentic materials such as court transcripts, online forums, or real-world investigative documents to improve realism and ecological validity?

---
(3) Lack of Technical Analysis:

The performance gap essentially reflects two underlying technical issues:
- insufficient safety alignment, since the model still tends to follow harmful or deceptive instructions; and
- limited long-context understanding, as detecting criminal or deceptive behavior often requires reasoning over extended context.
It would be helpful if the authors could analyze these aspects more deeply to clarify the technical reasons behind the observed gap.

---

> ### Author Response · Authors · 2025-11-21
> **Response to Reviewer sCDD**
>
> Thank you for your insightful comments. Here are our responses:
>
> ## (1) Nature of the Performance Gap
> First, as stated in the paper, our primary goal is to ***"highlight a potential risk-amplification effect: LLMs may be more exploitable for facilitating crime than for preventing it"***(lines 88-90). The focus is not whether the model has an inherent deficiency but whether its behavior shows an asymmetric vulnerability in realistic deployments.
>
> Second, our findings are designed not to be confounded by task-intrinsic difficulty. To control scenario complexity, we constructed a balanced dataset containing both criminal-success and detective-success scenes (Appendix B.1). These two narrative types differ in difficulty. Taking the detective agent as an example, detective-success narratives such as *The Invisible Guest* provide clear factual clues, for instance *"Amy's husband is only a junior bank clerk, whereas the suspect has extensive banking connections,"* which make it easier to identify moral disengagement in claims like *"Amy transferred assets with her husband's help."* In contrast, criminal-success narratives give the detective ambiguous information, such as incomplete key timelines or partially aligned witness accounts, making it substantially more challenging for the model to determine whether a suspect's statements contain deception.
> We further analyzed CTAR and OTDA across different narrative perspectives. As expected, the inherent characteristics of these narrative settings lead to minor variations. However, **the observed gap between criminal expression and detection remains consistently present across all difficulty settings**, implying that this discrepancy cannot be simply attributed to scenario difficulty.
>
> In addition, the research results are not driven by variation in contextual complexity across tasks. To control potential bias from context length, we examined context token-length distributions across tasks. The average length in criminal detection tasks exceeds that in criminal expression tasks by only 2.90%, indicating that the performance gap cannot be attributed to differences in contextual length.
>
> The above details are updated to Appendix F.3.1 and Appendix F.3.3 in the revised version.
>
> ## (2) Use of Film-Based Scenarios
> As **the first systematic study of the asymmetry between criminal expression and criminal detection in LLMs**, we selected films in order to conduct experiments under controlled and repeatable conditions. As stated in the paper (lines 123-125),  film narratives provide coherent causal chains and repeatable multi-agent interactions, enabling balanced scenario pairing and controlled rewriting.
>
> Besides, **incorporating real-world sources such as court transcripts, investigative reports, and forum discussions to improve realism and ecological validity can serve as a meaningful direction for future work**. However, these sources often involve legal or ethical restrictions and frequently lack the consistent logical structure needed for systematic comparisons. Many case materials rely on conjecture rather than verified reasoning, and building a high-quality corpus would require extensive anonymization, formal clearance, and substantial preprocessing to ensure comparability. Besides, we also positioned this direction as future work (lines 458-460).
>
> ## (3)  Lack of Technical Analysis
> Our study primarily reveals asymmetric capability patterns in LLM behavior when deployed in realistic settings. Based on this understanding, we further analyze the technical factors underlying this performance gap as follows:
>
> (i) Safety Alignment: **We find that insufficient safety alignment is a plausible factor behind models' tendency to follow harmful or deceptive instructions.** For GPT-4o, which adopts stronger alignment measures, we observe that in scenarios with clear criminal implications it often immediately provides normative responses such as *"Considering ethical principles, I cannot conceal this incident."* In less explicit cases, GPT-4o gradually self-adjusts over multiple turns and eventually offers morally guided suggestions, such as *"I would prefer to admit responsibility and seek help."* Therefore, we view the next key step as developing alignment strategies that explicitly target this asymmetric vulnerability.
>
> (ii) Long-Context Understanding: **We find that limited long-context understanding is not the main cause of the performance gap.** CTAR and OTDA remain stable across input-length intervals, and longer contexts neither increase criminal traits expression nor improve detection. Comparing Qwen3-32B with the much longer-context Qwen3-235B yields the same asymmetry between expression and detection. This indicates that the gap stems not from context limitations but from a deeper imbalance in the models' abilities to express versus detect criminal traits.
>
> The above discussion is updated to Appendix F.3 in the revised version.

---

### Meta-Review · Area_Chair_hkfP · 2025-12-24

**Summary:**

The paper introduces PRISON, a tri-perspective benchmark that evaluates the criminal potential and the ability to detect such harmful responses of LLMs along five dimensions. Through extensive experiments on various open-sourced and commercial LLMs, the paper reveals the risk-amplification asymmetry that models tend to make criminal expressions while they are not sufficiently capable of recognizing these criminal contents.

The authors have addressed most concerns during the rebuttal. However, there still exists concern about the experimental setting, and I feel the criminal-perspective in the framework seems like a sophisticated role-based jailbreak and remain unclear about the real-world harmfulness of the revealed asymmetry (which is claimed as the core contribution).

Nevertheless, given that most reviewers stay positive towards this paper, my recommendation is to accept.

**Reviewer Concerns:**

The reviewers raised the following concerns during rebuttal.

- sCDD: task difficulty gap between criminal potential and detection; dramatic scenarios; lack of technical analysis.
- k64m: lacks detailed elaboration on the detective’s task; “God” perspective validation issue; dramatic plots in the tested scenarios.
- Wu5t: artificial framework and lacks details on specific responses to determine if and how they are harmful; missing baseline of detective task; limited diversity of scenarios, only based on 10 movies; no larger reasoning models are evaluated (especially for the detective task with less useful information and context).

During the rebuttal, the authors have provided discussion and clarification to address concerns from reviewers sCDD and k64m. As for reviewer Wu5T's concerns, the authors conducted additional manual evaluation on the detective task and experiments on large reasoning model, and reviewer Wu5T acknowledged that most concerns were addressed, but the reviewer is still not fully convinced regarding the practical usability of the framework.

**Reviewer Scores:**

The reviewers would keep their initial scores.

---

### Decision · Program_Chairs · 2026-01-26

Accept (Poster)